# Behavioral Modeling of DC/DC Converters in Self-Powered Sensor Systems with Modelica

**DOI:** 10.3390/s21134599

**Published:** 2021-07-05

**Authors:** Jan Kokert, Leonhard M. Reindl, Stefan J. Rupitsch

**Affiliations:** Chair for Electrical Instrumentation and Embedded Systems, Department of Microsystems Engineering—IMTEK, University of Freiburg, Georges-Köhler-Allee 103, 79110 Freiburg, Germany; reindl@imtek.uni-freiburg.de (L.M.R.); stefan.rupitsch@imtek.uni-freiburg.de (S.J.R.)

**Keywords:** DC/DC converter, behavioral modeling, long-term simulation, wireless sensor nodes, efficiency function, energy harvesting, power management ICs

## Abstract

DC/DC converters are the essential component of power management in applications such as self-powered systems. Their simulation plays an important role in the configuration, analysis and design. A major drawback is the lack of behavioral models for DC/DC converters for long-term simulations (days or months). Available models are cycle-to-cycle-based due to the switch-mode nature of the converters and are therefore not applicable. In this work, we present a new behavioral model of a DC/DC power converter. The model is based on a thorough discussion of the model aspects that are relevant for self-powered systems, such as electrical representation and the causal connection if input and output. The model implementation is shown in the Modelica language and is available as an open-source library. The highlights of the model are a feedback controller for operation at the maximum power point (MPP), a loss-based efficiency function, and the start/stop behavior. The model’s capabilities are demonstrated in a 24h-experiment to predict voltage levels and the conversion efficiency.

## 1. Introduction

The term “self-powered systems” describes systems that are electrically separated from their environment and contain at least one energy-harvesting source and an energy storage. Our target applications are low-power systems such as wireless sensor nodes (WSN) powered by energy harvesting (EH) [1]. Here, systems should be small and inexpensive [2] and are often installed in large quantities [3]. The system needs to have an efficient energy extraction to deliver sufficient power to the consumers similar to sensors and actuators. Furthermore, it needs to cope with varying input power due to environmental conditions and manage an energy storage. Consequently, the system needs a sophisticated power management [4]. The essential component in a power management are DC/DC converters, which transfer power and adjust voltage and current levels.

### 1.1. Simulation: Level of Detail vs. Time Scale

Modeling and simulation of systems in general is important to not undergo a lengthy try-and-error approach during the design phase. Simulations and their results can be differentiated by the level of detail and the captured time scale. In the context of designing DC/DC converters, a high level of detail and a short time scale (<1 s) is applied to check the efficiency, the control stability and the electromagnetic compatibility (EMC). Contrarily, in the context of self-powered system design, a low level of detail and a long time scale (>1 h) is applied to check the dimensioning of the storage size and the duty cycles of the electrical consumers.

Our research is focused on long-term experiments and energy balance modeling of self-powered wireless sensor systems [5]. We consider the DC/DC converters to be self-contained building blocks in which detailed short-term effects are already captured by a proper circuit design [6].

### 1.2. Cycle-to-Cycle vs. Behavioral Modeling

Modern power converters are switch-mode devices operating at switching frequencies in the range of 100 kHz to 3 MHz. The usual approach of modeling a power converter is to simulate every cycle of charging and discharging the internal storage elements similar to inductors and capacitors. Such a cycle-to-cycle simulation can be carried out by tools such as LTSpice or TINA-TI. The simulation is very detailed but the required calculation duration $t_{\mathrm{calc}}$ is huge regarding the simulated time span $t_{\mathrm{sim}}$.

If weeks or even years shall be simulated, this leads to an unmanageable computational effort and such a high level of detail is not necessary. The proposed model describes the steady-state behavior of a DC/DC converter, which means that internal switching is not modeled but averaged. Our model is written in the Modelica language, a hierarchical, multi-physical and non-causal equation-based modeling language. The term “non-causal equation-based” means that modeling can be done using DAEs (differential algebraic equations). The solver performs symbolic variable transformation so that cause and effect of a certain behavior do not have to be determined. Hierarchical modeling means that the DAEs can be structured in blocks and displayed graphically.

### 1.3. Power Class of Self-Powered Sensor Systems

Besides the time scale, the power throughput of the system determines, which effects are relevant for the converter model. Sensor systems are typically low-power devices and converters operate in a range from 1 mW to 10 W. In the field of renewable energies and microgrid, typical powers are in the range from 100 W to 10 kW, which is about six orders of magnitude greater.

For EH-systems, the focus is on reducing standby power. Furthermore, a very low start-up voltage is desirable to avoid non-conversion losses. Maximum power point tracking (MPPT) is often accomplished by the fractional open-circuit voltage method (FOCV) [7]. For mid-power systems, the focus is on a high conversion efficiency and there is enough energy available for FPGAs implementing sophisticated algorithms such as the perturb and observe method (P&O) [8,9].

## 2. Fundamentals of Self-Powered Systems and Power Converters

### 2.1. Structure and Functions a Micro Power Management

The principle structure of self-powered systems is shown in Figure 1 and consists of three main components: An energy harvester, a storage element and electrical consumers such as microcontrollers and sensors. To connect these components, two power converters are used in between. There are three major functions of a power management illustrated with blue curly braces and explained next:**Energy extraction:** The output impedance of the harvester varies with the input power due to the environmental fluctuations. To extract as much energy from a harvester as possible, a continuous matching called maximum power point tracking (MPPT) is needed. The first power converter in Figure 1 is typically a step-up converter. For instance, a single solar cell only offers a voltage of about 0.5 V at the MPP, but a Li-Ion battery has an open-circuit voltage of 3.7 V.**Storage interaction (charge and protect):** Most secondary batteries (e.g., Li-ion) need a constant-current, constant-voltage charging scheme (CCCV). Due to power fluctuations, this scheme cannot be followed completely, but at least a voltage limiting to prevent under- and over-charge must be implemented in both converters.**Voltage supply:** A regulated constant voltage is required to supply the consumers. The second power converter is typically a step-down converter, as e.g., a low-power microcontroller needs a stable voltage of 1.8 V, which is below the battery voltage.

### 2.2. Power Management Integrated Circuits (PMICs)

A more enhanced DC/DC converter is often referred to as PMIC (power management integrated circuit). Although the term PMIC refers to a wide range of chips and modules, most PMIC include a DC/DC converter or at least its control part. Besides the energy conversion process, additional features such as MPPT and protection circuits are integrated in an IC. These features make them ideal to fulfill the power management functions mentioned above. PMICs differ mainly in the structure (buck/boost) and the implemented feedback concept (see later in Section 4.4).

Designers of self-powered systems typically prefer commercially off-the-shelf (COTS) solutions due to cost, the availability of evaluation boards and fast board bring-up. There are several popular PMICs (currently around 20) for energy-harvesting WSNs that have an impact on system design, as overviews in [6,10] show.

### 2.3. Structure of PMIC vs. the Presented Model

The model presented in this work is driven by commercially available EH-PMICs. However, PMICs show a high structural variance from one type to another. Besides the DC/DC converter, every PMIC integrates a different subset of additional features such as MPPT, over/under voltage protection, additional battery backup, supercapacitor charge balancing and power-good indication.

It is difficult to capture this variance in one self-contained model. Instead, the model itself should be hierarchical and consist of the key component, a *DC/DC converter presented in this paper* and other components, which are then freely arranged.

## 3. Related Work

The **ModelicaStandardLibrary 4.0** [11] only contains switch-mode models of a DC/DC converters (e.g., Modelica.Electrical.PowerConverters.DCDC.ChopperStepDown), which is only mentioned here for the sake of completeness. Only the power transformation is modeled by ideal transistors and the PWM control is outsourced as another component (signalPWM).

**Torrey et al.** suggest in [12] a behavioral model of a DC/DC converter using Modelica. By means of a proportional–integral (PI) controller and a commanded output voltage the output current is controlled. The corresponding load is then reflected to the input as a current sink by dividing $P_{\mathrm{out}}$ by $V_{\mathrm{in}}$ and a constant efficiency value. Another feature is a minimum operating input voltage $V_{\min}$, where the input current sink is set to zero for too low input voltages. The converter efficiency is modeled as a constant value and an additional input resistor representing quiescent currents. The model is intended to be used for high power applications (approx. 5000 W).

The open-source **PhotoVoltaics Modelica library** [13] includes a DC/DC converter to extract energy from solar cells (PhotoVoltaics.Components.Converters.DCConverter). The input is modeled as voltage source, which is commanded externally by a MPP tracker, following the P&O concept. The input power is calculated and compared to the output power by means of an integral (I) controller. The controller controls the variable current source, which results in an output voltage that cannot be limited in any way. The model does not consider any losses. Furthermore, the converter is continuously running, and start-up or shutdown behavior is modeled.

The commercial **EDrives library** [14] includes three inverter models (DC/AC converters) to provide power to electric drives. The EDrives.PowerConverters.Averaging model neglects switching effects. The output is represented by a voltage source, which is commanded externally. The output power is calculated by multiplying the commanded $V_{\mathrm{out}}$ and the resulting $I_{\mathrm{out}}$, which is then reflected to the input. An I controller compares $P_{\mathrm{in}}$ and $P_{\mathrm{out}}$ and controls the input current sink. The model assumes a constant efficiency of 100% and runs continuously.

**Oliver et al.** propose in [15] a behavioral model of a multi-output DC/DC converter. The input is represented by current sink and the outputs are represented by voltage sources with internal resistances. The focus is on modeling transient behaviors (start-up, load steps and output cross-over effects) with RCL networks without simulating the switching of the converter. A state-machine handles remote on/off an protections.

**Behrmann et al.** present in [16] a toolbox for energy analysis of self-powered sensors written in MATLAB Simulink. The toolbox features several blocks, which communicate via power ports. Thus, voltage and current levels are not considered and start-up and MPPT cannot be analyzed. However, the converter efficiency is modeled as a look-up table.

In summary, it can be stated that the related work only covers basic behaviors of the converter and there is no model available that is specifically for low-power applications.

## 4. Model Scope and Discussion Based upon Modeling Aspects

In this section, we define the scope of our DC/DC converter model. All relevant aspects are highlighted, and the methodology of the implementation is discussed. Furthermore, the considerations are compared to relevant approaches from literature.

### 4.1. Electrical Representation

The electrical representation of the converter’s input and output is important for the interaction with other components such as the energy harvesters and the loads. For the input, a passive component (resistor or conductor) is preferable, since voltage and current levels are determined by the connected harvester. Still, active components are used as input in the literature [15,17], which can lead to negative input voltages. A conductor is beneficial compared to a resistor, as it can be disabled completely with a modest numerical value of zero ($G_{\mathrm{in}}=0$ S), whereas for the resistor the value infinity ($R_{\mathrm{in}}=\infty \;\Omega$) is required. Furthermore, a resistivity of zero is not needed in reality.

The converter output must be an active element (voltage or current source) that can be converted by means of an additional resistor according to the Norton-Thevenin theorem. In our model, the output of the DC/DC converter is represented by a controlled current source. This represents very well the entirety of pulse currents from the inductor and a filter network of a real DC/DC converter.

### 4.2. Causal Connection of Input and Output

After defining the input and output ports, the causality between them needs to be established. There are two possible approaches: (1) a **forward definition** (approach of energy availability), where the power into the converter is controlled which then results in a certain output [13] or, (2) a **backward definition** (approach of energy demand), where the output is controlled, which results in a certain power draw at the input of the converter [12,14,15].

The first approach will be recommendable, when the input energy is limited and aspects such as MPPT should be implemented in the model. The second approach will be preferable if unlimited energy input can be assumed and if the load supply should be modeled precisely, including the load regulation.

Our model implements the forward definition, as input energy is limited. Only power that is consumed at the input is accessible at the output, which is further described in Section 5.3. Another reason is that this corresponds to the flow of energy in reality. Here, the input energy is first stored by an inductance and a PWM-controlled switch and is then accessible at the output capacitor.

### 4.3. Efficiency Function and Power Losses

Self-powered systems typically only have limited access to energy and thus, the efficiency of the components becomes crucial. The efficiency η of a power converter is the ratio of instantaneous output power $P_{\mathrm{out}}$ and input power $P_{\mathrm{in}}$ according to
(1)$$\begin{array}{cc} \eta & =\frac{P_{\mathrm{out}}}{P_{\mathrm{in}}}=\frac{V_{\mathrm{in}}\;\cdot \;I_{\mathrm{in}}}{V_{\mathrm{out}}\;\cdot \;I_{\mathrm{out}}}=f\left(V_{\mathrm{in}},I_{\mathrm{in}},V_{\mathrm{out}},I_{\mathrm{out}}\right). \end{array}$$

Due to the converter’s working principle, the four port values (input voltage $V_{\mathrm{in}}$, input current $I_{\mathrm{in}}$, output voltage $V_{\mathrm{out}}$ and output current $I_{\mathrm{out}}$) correlate with each other. For example, if the input voltage drops, the input current will increase if a constant load is connected. With a real power converter, this correlation is non-linear and η is not simply a constant value. Instead, the efficiency should be considered to be a function depending on at least one of the four port values. Unfortunately, other available models only offer to specify a constant value for the efficiency [12] or even assume an efficiency of 100% [13,14]. The next best approach is to provide a fit function [18] or a look-up-table [15,16] for the efficiency. This may be practical, but it is not very accurate from a physical point of view.

When the efficiency function is modeled, one must keep in mind that it is just a calculated and unitless ratio and has no direct physical equivalent. However, the efficiency can be explained by introducing the concept of power losses, which also leads to a deeper understanding of the converter. Power losses originate from capacative switching losses, the switch resistance and parasitic resistance in the inductor and capacitors [18,19]. The total power losses of all components $P_{\mathrm{loss},\;\mathrm{tot}}$ reduces the output power and thus the efficiency according to
(2)$$\begin{array}{cc} \eta & =\frac{P_{\mathrm{in}}-P_{\mathrm{loss},\;\mathrm{tot}}}{P_{\mathrm{in}}}=1-\frac{P_{\mathrm{loss},\;\mathrm{tot}}}{V_{\mathrm{in}}\;\cdot \;I_{\mathrm{in}}}. \end{array}$$

In this paper, we do not consider the internal structure of the parasitic components for three reasons: (1) The values of the internal parasitic components such as on-resistance and capacitances are not known and cannot extracted completely from the datasheet. (2) The individual loss terms can be quite complicated (see [19]) and simple models are usually preferable. (3) The equations are sectioned depending on the converter type (buck or boost) and operation mode (continuous conduction or discontinuous conduction). Our approach is to observe the efficiency curves and choose four simple loss terms with respect to $V_{\mathrm{in}}$ and $I_{\mathrm{in}}$, which is further described in Section 5.4.

### 4.4. Feedback Control

Off-the-shelf power converters are active components with a closed-loop control. A behavioral model should also implement a feedback concept for the following reasons: (1) The function of a DC/DC converter in the power management is determined by the feedback concept (see Section 2.1). For **energy extraction**, the input voltage $V_{\mathrm{in}}$ is controlled by the MPP tracking algorithm (FOCV or P&O). For **storage interaction** with CCCV charging, a combination of $I_{\mathrm{out}}$ and $V_{\mathrm{out}}$ is used. And for **voltage supply**, the output voltage $V_{\mathrm{out}}$ is controlled. (2) The converter model is embedded into a complete system of energy sources and sinks. The above voltages and currents are setpoints that can deviate from the actual value. They are no fixed values, since this would lead to contradictions within the system model.

A feedback control monitoring $V_{\mathrm{out}}$ is implemented in [12] and MPPT (monitoring $V_{\mathrm{in}}$) is implemented in [13]. Our proposed model features setpoints for all power management functions, which is further described in Section 5.7. Both buck and boost converters are modeled in one approach.

### 4.5. Converter Start-Up and Shutdown

The start-up and shutdown behavior is another relevant behavior of a DC/DC converter, especially in energy-harvesting applications where the input voltages are small (< 1 V). Small amounts of energy are then available, but cannot be used because the input voltage is below the minimum working voltage $V_{\min}$ and the converter is switched off. Another phenomenon is the minimum cold-start voltage $V_{\mathrm{start}}$. Its value is higher than $V_{\min}$ and must be exceeded if the converter has not been running before (it was *cold*). After successful start-up, the converter will provide power to internal circuitry to reduce the minimal working voltage. The difference of these voltages can be quite significant, e.g., consider the BQ25570 where $V_{\min}$ = 100 mV and $V_{\mathrm{start}}$ = 600 mV [20].

Assuming a constantly running converter overestimates the available energy for the system loads. Only a few other publications model the minimum working voltage [12,15]. If the cold start is not implemented, simulation loops can occur by continuously switching on and off the converter. Our model implements both the minimum working voltage and minimum cold-start voltage by means of a state machine, which is further described in Section 5.6.

## 5. The Proposed Behavioral Model

Our model is part of an open-source library [21] to simulate EH-powered WSNs. The structure of the proposed model is shown in Figure 2. As we use Modelica, it is easy to create hierarchical models and make use of existing components from the ModelicaStandardLibrary 4.0 [11]. The model components are explained in the following.

### 5.1. Overview of the Power Path

In the following Section 5.2, Section 5.3 and Section 5.4, we will explain the power path of the converter. First, we will present the electrical representation of output and input. Then we will explain how to determine the output current. Lastly, we will explain how the efficiency within this path is modeled. The three corresponding icons are shown in Figure 3.

### 5.2. Electrical Representation and Working Principle

We model the input of the DC/DC converter by the controlled conductor $G_{\mathrm{in}}$. The output is represented by a controlled current source $I_{\mathrm{out}}$. The input conductivity $G_{\mathrm{in}}$ will consume the input power $P_{\mathrm{in}}$, which is adjusted by a PI controller and a start-stop logic. The converter efficiency η is modeled by means of internal power losses $P_{\mathrm{loss},\;\mathrm{tot}}$. The output power $P_{\mathrm{out}}$ is then derived from $P_{\mathrm{in}}$ by subtracting these internal losses. Finally, the output current $I_{\mathrm{out}}$ is calculated by dividing the output power $P_{\mathrm{out}}$ by the output voltage $V_{\mathrm{out}}$ according to
(3)$$\begin{array}{cc} I_{\mathrm{out}} & =\frac{P_{\mathrm{out}}}{V_{\mathrm{out}}}=\frac{P_{\mathrm{in}}\;\cdot \;\eta }{V_{\mathrm{out}}}. \end{array}$$

### 5.3. Determination of the Output Current

Taking a closer look at (Equation 3) reveals that there is a risk of a division by zero, which can be problematic during simulation. The condition where the denominator is zero ($V_{\mathrm{out}}=0$) will occur during start-up, when the converter outputs no voltage but consumes a certain start-up input power. Furthermore, a short circuit at the output equals $V_{\mathrm{out}}=0$.

To solve the division-by-zero issue, we substitute $V_{\mathrm{out}}$ in (Equation 3) by $V_{\mathrm{out}}^{\prime }$ as defined in (Equation 4). An exponential term is added to $V_{\mathrm{out}}$, which evaluates to 0.1 V for $V_{\mathrm{out}}$ = 0 and then rapidly decays to zero for larger output voltages, as shown in Figure 4. For a regular output voltage (e.g., 3.3 V), the additional term evaluates to $4.7\times 10^{-16}$ V and is then negligible.
(4)$$\begin{array}{cc} V_{\mathrm{out}}^{\prime } & :=V_{\mathrm{out}}+\frac{e^{-10\;\cdot \;V_{\mathrm{out}}}}{10} \end{array}$$

The authors in [12] faced a similar division-by-zero problem and solved it by adding a small constant ($10^{-10}$) to the denominator. However, this can result in a very high output current at start-up due to the reciprocal operation. e.g., for an output power of 1 W, (Equation 3) yields an output current of $1\times 10^{10}$ A for $V_{\mathrm{out}}=1\times 10^{-10}$ V.

### 5.4. Efficiency Calculation Based on Power Losses

We model the converter efficiency by first observing the main characteristics of the efficiency curves. In this paper, we focus on step-up (boost) converters, but the principle is also applicable for step-down converters. For step-up converters, the relevant curves, provided by the manufacturer, are the efficiency η plotted against $I_{\mathrm{in}}$ and $V_{\mathrm{in}}$ at a constant $V_{\mathrm{out}}$. For step-up converter, the efficiency is usually shown w.r.t $I_{\mathrm{out}}$ and $V_{\mathrm{in}}$. Figure 5 shows in solid lines the efficiency curves of the ADP5090 boost converter, extracted from the original datasheet [22] for $V_{\mathrm{out}}=3.0V$. The raw measurement data and a script to obtain the simulated results is available here: [21]. The dashed lines show the simulated efficiency according to our model. The used formulas and parameters are explained on the next page. However, first, we study the general behavior of the graphs, where we observed the following four characteristics:

The input current sweep shows a plateau with a flat maximum (i.e., ≈ 1 mA for the ADP5090), whereas the input voltage sweep is monotonically increasing without a clear maximum.The curves in the current sweep show a similar shape, but the higher Vin, the higher the efficiency. This can be explained by the fact that the $V_{\mathrm{out}}$-to-$V_{\mathrm{in}}$ ratio moves closer to 1, which causes fewer losses.In both graphs, the efficiency drops rapidly for very small input powers. The behavior is similar even if $V_{\mathrm{in}}$ and $I_{\mathrm{in}}$ are swept independently of each other.Only in the $I_{\mathrm{in}}$ sweep, the efficiency decreases for high input currents. It will be especially dominant if the input voltage is small. There is no comparable slope at the ”right side” in the $V_{\mathrm{in}}$ graph.

As motivated previously, our approach is to choose appropriate loss terms $P_{\mathrm{loss},i}$, which are a function of $V_{\mathrm{in}}$ and $I_{\mathrm{in}}$. This differs from other approaches, in which the efficiency is described directly by a fit function [18]. Finally, the sum of all power losses $P_{\mathrm{loss},\;\mathrm{tot}}$ is calculated in (Equation 5) and is then subtracted from the input power in $P_{\mathrm{in}}$ (6). During the whole simulation, the condition $0<P_{\mathrm{loss}}<P_{\mathrm{in}}$ must hold true. Otherwise, the efficiency becomes negative of greater than 100%, which can happen if a fit parameter is not correctly chosen. As a precaution, we added the max function to (6).
(5)$$\begin{array}{cc} P_{\mathrm{loss},}\;\mathrm{tot} & =\sum_{i=1}^{n}P_{\mathrm{loss},i} \end{array}$$
(6)$$\begin{array}{cc} P_{\mathrm{out}} & =\max(0,P_{\mathrm{in}}-P_{\mathrm{loss},\;\mathrm{tot}}) \end{array}$$

Generally, the individual loss terms $P_{\mathrm{loss},i}$ can be arbitrary functions in $V_{\mathrm{in}}$ and $I_{\mathrm{in}}$. However, we want a dedicated control of the curves in either the $I_{\mathrm{in}}$ sweep or the $V_{\mathrm{in}}$ sweep and take advantage of the fact that the fraction in the efficiency function (Equation 2) contains $V_{\mathrm{in}}\;\cdot \;I_{\mathrm{in}}$. In doing so, the fraction can be shortened, and we obtain loss terms, which act only in the $V_{\mathrm{in}}$ sweep or the $I_{\mathrm{in}}$ sweep. The four loss terms are described by (Equation 7) to (10) and are explained in the next sections. The fit parameters $k_{1}$ to $k_{4}$ of the loss terms can be extracted directly from the characteristic curves. The actual values for the plot in Figure 5 (ADP5090 converter) are: $k_{1}=0.01$ V, $k_{2}=0.11\;\sqrt{V}$, $k_{3}=1.2$ μW and $k_{4}=1.35$ Ω. These values can change depending on the buck/boost converter chip.
(7)$$\begin{array}{cccc} P_{\mathrm{loss},}\;1 & =k_{1}\;\cdot \;I_{\mathrm{in}} & & \;\to \eta =1-\frac{k_{1}}{V_{\mathrm{in}}} \end{array}$$
(8)$$\begin{array}{cccc} P_{\mathrm{loss},}\;2 & =k_{2}\;\cdot \;I_{\mathrm{in}}\;\cdot \;\sqrt{V_{\mathrm{in}}} & & \;\to \eta =1-\frac{k_{2}}{\sqrt{V_{\mathrm{in}}}} \end{array}$$
(9)$$\begin{array}{cccc} P_{\mathrm{loss},}\;3 & =k_{3} & & \;\to \eta =1-\frac{k_{3}}{V_{\mathrm{in}}\;\cdot \;I_{\mathrm{in}}} \end{array}$$
(10)$$\begin{array}{cccc} P_{\mathrm{loss},}\;4 & =k_{4}\;\cdot \;I_{\mathrm{in}}^{2} & & \;\to \eta =1-\frac{k_{4}\;\cdot \;I_{\mathrm{in}}}{V_{\mathrm{in}}} \end{array}$$

#### 5.4.1. Power Loss Proportional to $I_{\mathrm{in}}$

The first loss term (Equation 7) is proportional to $I_{\mathrm{in}}$. In the efficiency function, $I_{\mathrm{in}}$ cancels out and we obtain horizontal lines in the $I_{\mathrm{in}}$ sweep. This also represents the maximum efficiency and the plateau. Furthermore, we obtain the characteristic $\frac{1}{x}$ shape in the $V_{\mathrm{in}}$ sweep, where the efficiency increases monotonically with increasing $V_{\mathrm{in}}$. Already this simple term models the first two behaviors that we observed at the beginning.

#### 5.4.2. Power Loss Proportional to $I_{\mathrm{in}}\;\cdot \;\sqrt{V_{\mathrm{in}}}$


The second loss term (8) builds on (Equation 7), by adding further losses which are proportional to the square root of $V_{\mathrm{in}}$. This term is a step towards defining a global maximum efficiency. For a moment, consider the loss term $P_{\mathrm{loss}}=k_{2}\;\cdot \;I_{\mathrm{in}}\;\cdot \;V_{\mathrm{in}}$ that will lead to a constant efficiency, when plugged into (Equation 2) (resulting in $\eta_{\max}=1-k_{2}$). Given that defining an $\eta_{\max}$ is a too hard restriction, the $\sqrt{V_{\mathrm{in}}}$ is a good compromise. In summary, this term helps to push down the efficiency curves in the $I_{\mathrm{in}}$ sweep for decreasing $V_{\mathrm{in}}$.

#### 5.4.3. Constant Power Loss

The third term given by (9) describes power losses that are independent of the input power and are always present. This comprises losses such as quiescent currents (leakage) and internal bandgap references. It models the decreasing efficiency in the $I_{\mathrm{in}}$ sweep for low input currents (slope on the ”left side“). For very low input powers ($P_{\mathrm{in}}<k_{3}$, typically below 1 μW), the efficiency becomes negative. Fortunately, the model handles this situation by switching-off the converter and by (6).

#### 5.4.4. Power Loss Proportional to $I_{\mathrm{in}}^{2}$


The fourth loss term is given by (10) and models the decreasing efficiency in the $I_{\mathrm{in}}$ sweep for high input currents (slope on the ”right side”). The loss originates from by conduction losses, which dominate at high currents according to Ohm’s law ($P=R\;\cdot \;I^{2}$).

### 5.5. Overview of the Control Path

In the following Section 5.6, Section 5.7 and Section 5.8, we will explain the control path of the converter. First, the start-stop mechanism is presented. Afterwards, the determination of the feedback error and the PI controller is explained. The three corresponding icons are shown in Figure 6. In Section 5.9 the operation at the MPPT is explained, which is an iteration of all three components.

### 5.6. Minimum Start-Up and Working Voltage

Equations in Modelica are primarily intended to model continuous behavior, whereas algorithms are used to describe state machines. Therefore, we model the start-up and shutdown behavior as an algorithm, whereby the Boolean state variable isOn is introduced (see Figure 7). It indicates whether the converter is running or not.

To determine the transition between on and off, the input voltage $V_{\mathrm{in}}$ is compared to the minimum working voltage $V_{\min}$ and the cold-start voltage $V_{\mathrm{start}}$, respectively. In addition, the output setpoint voltage $V_{\mathrm{set}}$ is used: Consider a case in which an energy storage is connected to the converter, which is charged in such a way that its terminal voltage is higher than the intended output voltage of the converter. Then, the converter should not switch on even though the input condition is fulfilled ($V_{\mathrm{in}}>V_{\mathrm{start}}$).

### 5.7. Feedback Error Determination and Setpoints

A proportional–integral (PI) controller controls the input conductivity $G_{\mathrm{in}}$ to reach the following setpoints: At the input, the voltage $V_{\mathrm{mpp}}$ can be commanded to achieve MPPT (see Section 5.9). At the output, two setpoints (limits) can be provided: The maximum output voltage $V_{\mathrm{set}}$ and the maximum output current $I_{\mathrm{set}}$. By means of these parameters, a regulated voltage supply as well as a current-limiting storage charging (CCCV) can be implemented. The model is designed in such a way that the converter stays on for as long as possible. Especially at high loads, the converter will reduce the output power by neglecting the output setpoints to keep the converter alive ($V_{\mathrm{in}}>V_{\min}$).

In summary, three setpoints are monitored by the controller, which results in three inequations, given in (Equation 11). The controller will try to fulfill at least one of the inequations by equality. The total error of all setpoints $e_{\mathrm{PI}}$ is calculated using a minimum function (12). When the converter is off, the error $e_{\mathrm{PI}}$ is certainly >0. The connected PI controller would integrate this error, which leads to incorrect results when switched on again. Therefore, $e_{\mathrm{PI}}$ must be set to zero during the off-state.
(11)$$\begin{array}{cc} V_{\mathrm{in}} & \geq \max\left(V_{\mathrm{mpp}},V_{\min}\right);\;\;V_{\mathrm{out}}\leq V_{\mathrm{set}};\;\;I_{\mathrm{out}}\leq I_{\mathrm{set}} \end{array}$$
(12)$$\begin{array}{cc} e_{\mathrm{PI}} & =\left\{\begin{array}{cc} \min(V_{\mathrm{mpp}}-V_{\mathrm{in}},\;V_{\mathrm{set}}-V_{\mathrm{out}},\;I_{\mathrm{set}}-I_{\mathrm{out}}) & \mathrm{if}\;\;\mathrm{isOn}\\ 0 & \mathrm{otherwise}\;\; \end{array}\right. \end{array}$$

### 5.8. Closed-Loop Feedback by a PI Controller

The feedback error signal $e_{\mathrm{PI}}$ is fed into the PI controller. The I part was added to continually mitigate the P offset error. A D part was not added, because the PI implementation is already fast enough, and the solution is less complicated. Equation (Equation 13) describes the PI controller’s transfer function. The output $y_{\mathrm{PI}}$ is connected to the input conductivity $G_{\mathrm{in}}$ (see Figure 2) by means of the start-stop logic according to (14). Here, a switched-off converter corresponds to $G_{\mathrm{in}}=0$.
(13)$$\begin{array}{cc} y_{\mathrm{PI}} & =k_{\mathrm{FB}}\;\cdot \;\left(\int \;\frac{e_{\mathrm{PI}}}{T_{\mathrm{FB}}}\;dt+e_{\mathrm{PI}}\right) \end{array}$$
(14)$$\begin{array}{cc} G_{\mathrm{in}} & =\left\{\begin{array}{cc} y_{\mathrm{PI}} & \mathrm{if}\;\;\mathrm{isOn}\\ 0 & \mathrm{otherwise}\;\; \end{array}\right. \end{array}$$

In a real converter, the controller parameters $k_{\mathrm{FB}}$ and $T_{\mathrm{FB}}$ determine the ripple response and stability of the converter output. An approach to calculate $k_{\mathrm{FB}}$ and $T_{\mathrm{FB}}$ is presented in [12]. Here, a sinusoidal load and a resistor is connected, and the transfer function of the controller is then analyzed for high and low frequencies.

In our model, the main reason for the PI controller is that the model actually compiles in Modelica and does not lead to an algebraic loop. In doing so, the two controller parameters can be chosen in a wide range. We found that for both $k_{\mathrm{FB}}$ and $T_{\mathrm{FB}}$, a value of $1\;\times \;10^{-4}$ gives the best results.

### 5.9. Operation at the Maximum Power Point

The model offers to connect an externally controllable voltage $V_{\mathrm{mpp}}$, to allow the converter to operate at the MPP. The voltage $V_{\mathrm{mpp}}$ represents the lowest value for $V_{\mathrm{in}}$ at limited input power, which is different to [13], where $V_{\mathrm{in}}$ is forced to be $V_{\mathrm{mpp}}$. For a better understanding, consider a scenario with real power source (with internal resistance) and a load at the output: as the load increases, the input voltage $V_{\mathrm{in}}$ in the model will only decrease as low as $V_{\mathrm{mpp}}$, which represents the minimum suitable input voltage. A further reduction of $V_{\mathrm{in}}$ will not result in extracting more power from the harvester.

The voltage can be provided externally via the port extMPP or by the parameter fixedMPP and with useExternalMPP being disabled. To disable the MPP feature completely, the voltage $V_{\mathrm{mpp}}$ can be set to 0 V. To determine $V_{\mathrm{mpp}}$, several approaches are used in reality (FOCV and P&O) [7,8]. The different techniques are not part of this DC/DC converter model. However, the MPP tracking is implemented by means of an external tracker, which follows the FOCV method and a pilot cell. The setpoint $V_{\mathrm{mpp}}$ is calculated as the fraction of the open-circuit voltage $V_{\mathrm{oc}}$ according to
(15)$$\begin{array}{c} V_{\mathrm{mpp}}=k\;\cdot \;V_{\mathrm{oc}}\;|\;k=0.8\;\mathrm{for}\;\mathrm{solar}\;\mathrm{cells}. \end{array}$$

## 6. Simulation Setup

To demonstrate the usability and the behavioral aspects of the model, a simple use case was created. The setup is shown in Figure 8, consisting of a solar irradiation simulator, a solar cell, the DC/DC converter and a LiPo Battery. The second power converter (cf. Figure 1) was omitted, as this converter typically operates only in the constant-voltage mode and is less interesting. Furthermore, CCCV is already done by the first converter. The model source code of the setup is part of our online library [21], where the path is: EnergyHarvestingWSN.PowerConverter.UnitTests.SolarADP5090.

The solar simulator is the same as in our previous publication [5]. It describes a sunny morning and a cloudy afternoon by means of sinus functions. The peak irradiation is 1000 W/m${}^{2}$. The solar cell is a two-diode model and comparable with the model in [13]. The area *A* is set to 3 cm${}^{2}$ and the short-circuit current density Jsc is set to 37 mA/cm${}^{2}$, resulting in peak efficiency $\eta_{\mathrm{sol},p}=18\%$. By means of an internal pilot cell, the open-circuit voltage is measured and scaled by k=0.8 and is then available at the $V_{\mathrm{mpp}}$ port. The chosen DC/DC converter is the ADP5090 from Analog Devices. As explained in Section 2.3, the MPPT is not implemented in this block. We set $V_{\mathrm{start}}$ to 0.5 V for this experiment. The battery model is based on [23] and shows the typical non-linear charging characteristics. The capacity is 50 mAh and the charge-stop voltage is 4.2 V.

The simulated timespan was half a day (43,200 s) and the simulation was carried out on a PC with an Intel^®^ Core^™^ i7 running at 2.5 GHz. As numerical solver, the program Open-Modelica 1.18.0-dev was used and an interpolation interval of 100 s was chosen.

## 7. Results

### 7.1. Simulation Performance

The compilation of the model takes approx. 10 s. After that, the actual required computational time $t_{\mathrm{calc}}$ (for the DASSL integration method) is below 1 s. Prolonging $t_{\mathrm{sim}}$ to one year (31,536,000 s), results in a $t_{\mathrm{calc}}$ = 1 min and a result file of approx. 200 MByte, which is still manageable. And still, the results are very detailed, and all voltage and currents are simulated precisely, as shown in Figure 9.

### 7.2. Behavior Discussion

Figure 9 (top) shows $V_{\mathrm{in}}$ in blue, $V_{\mathrm{mpp}}$ in green and $I_{\mathrm{in}}$ in red color. At t=6 h, the solar irradiation starts and $V_{\mathrm{in}}$ is increasing rapidly due to the non-linear IV-characteristic of the solar cell. At the times $t=\left[6.09\;h,13.03\;h,14.63\;h\right]$, the converter starts up. It looks as if the input voltage has overshoots at these points, but actually it is only because the converter needs at least an input voltage of 0.5 V to work. In off-state, no current is drawn from the solar cell, but when switched on, the converter regulates $V_{\mathrm{in}}$ down to $V_{\mathrm{mpp}}$. At t=15.47 h, the battery is full and no more current flows out of the converter into the battery. Consequently, the MPPT is stopped and $V_{\mathrm{in}}$ jumps from below 0.5 V to 0.6 V, which is the $V_{\mathrm{oc}}$ at this time.

Figure 9 (bottom) shows the DC/DC converter efficiency in magenta and the efficiency of the solar cell in yellow. The converter shows a peak efficiency of 80 % because $V_{\mathrm{in}}$ is always below 0.5 V during operation. Whenever the solar efficiency is high due to a high solar irradiation, the converter efficiency drops to as low as 55,4 % because of conduction losses ($P_{\mathrm{loss},\;4}$).

Additionally in Figure 9 (bottom), the terminal voltage of the LiPo battery (=$V_{\mathrm{out}}$) is shown in cyan. At t=15.47 h, the battery is full. The constant-current phase of a CCCV charging scheme cannot be followed, due to the environmental fluctuations. However, the charge stop at $V_{\mathrm{out}}=4.2$ V is visible.

## 8. Discussion

In the following, our work is discussed with respect to the start-of-the-art. In model scope (see Section 4), five aspects of a DC/DC converter model were highlighted. Table 1 shows how well the models perform on these criteria. It must be mentioned that none of the state-of-the-art models was specifically designed for self-powered or energy-harvesting systems. It is, therefore, not surprising that no model implements a separate cold-start voltage $V_{\mathrm{start}}$. Most of the publications use some kind of closed-loop controller; however only one commanded variable is available then. It is also noticeable that the best approach to model the efficiency is a simple look-up table. A variable efficiency, such as in our model, was not considered by any other publication.

## 9. Conclusions

This paper presented a new behavioral model for a DC/DC converter. First, a thorough discussion about the model scope and aspects for self-powered systems were given. Afterwards, the model with all relevant equations was explained and the graphical representation in Modelica was shown. A simulation setup and the results round off the paper. As discussed in Section 8, our model outperforms the other publication in the following points:The model implements a complete start-stop behavior with minimum working voltage $V_{\min}$ and the cold-start voltage $V_{\mathrm{start}}$.The converter efficiency is modeled as a function, which is based on losses and depends on $V_{\mathrm{in}}$ and $I_{\mathrm{in}}$. The loss terms are carefully selected to easily extract the parameters from the manufacturer’s datasheet.The closed-loop controller allows three modes of operation: CV (constant-voltage output), CC (constant-current output) and MPP (following the maximum power point by regulating Vin).

Our main contribution is that detailed simulations of energy-harvesting systems can be carried out now within seconds. This is supported by our demonstration setup. Here all relevant behavioral aspects are visible in the voltage and efficiency curves. The model helps to design a real system, by studying all losses and side-effects of the power conversion in a micro sensor system. The risk of a system failure due to energy shortage can be simulated without time-consuming field tests.

All parameters of our model can be extracted from the datasheet without knowing the internal structure of the converter. The published library (available online on GitHub [21]) provides datasets for popular PMICs such as the ADP5090 and the BQ25570 and a Python script to fit the parameters.

The language Modelica offers great features as explained in Section 1.2 and is the perfect tool for us: Writing down the equations makes the source code well readable, and the causal connection of elements does not have to be considered (such as in MATLAB Simulink). Furthermore, as the model is equation-based, it can be ported to other circuit simulators such as PSpice. Lastly, it is a non-commercial language and there are currently over 100 free libraries available [24].

In conjunction with other Modelica libraries, also other harvester types can be simulated, e.g., thermoelectric generators with [25]. For electromagnetic or piezoelectric energy harvesters, first a rectifier stage and a buffer capacitor is needed. Our future research steps are the simulation of real systems that are set up with our modular test platform [6]. Since Modelica also supports the simulation of sequential programs, dynamic duty cycling algorithms of the microcontroller shall be investigated and simulated. Finally, by simulating the energy balance, the reliability of self-powered systems shall be calculated and evaluated.

## Figures and Tables

**Figure 1 sensors-21-04599-f001:**
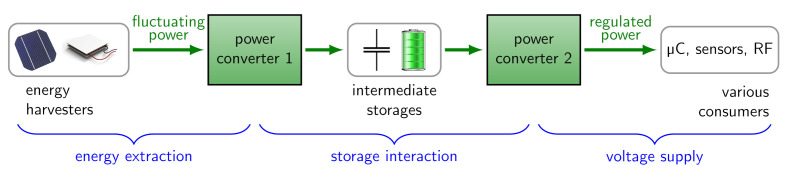
Power converters supply power to consumers from energy-harvesting sources.

**Figure 2 sensors-21-04599-f002:**
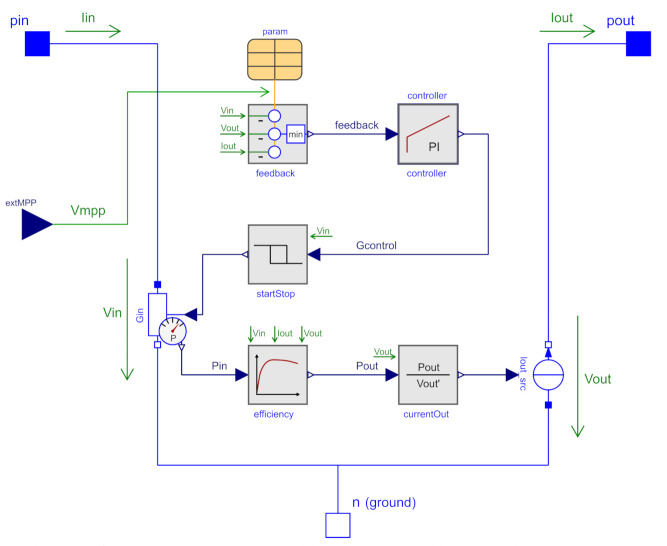
Internal structure of the proposed power converter model implemented in Modelica. The model uses components such as a conductor and a current source and new components depicted as gray blocks. The green labels are global variables that are available in all components. The blocks itself contain only equations and no further Modelica components.

**Figure 3 sensors-21-04599-f003:**
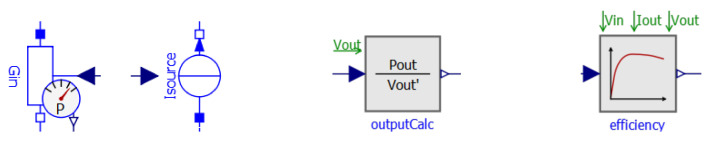
Icons of the components in the power path: input and output representation (**left**), determination of output current (**middle**) and efficiency calculation (**right**).

**Figure 4 sensors-21-04599-f004:**
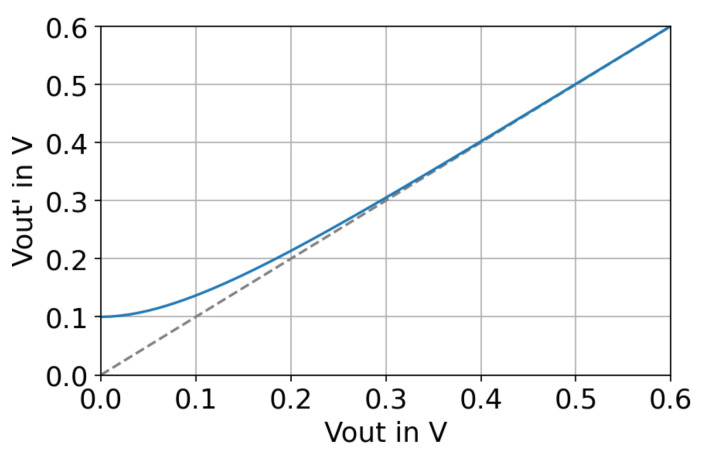
Substitution of $V_{\mathrm{out}}$ with $V_{\mathrm{out}}^{\prime }$ by adding an exponential decaying term.

**Figure 5 sensors-21-04599-f005:**
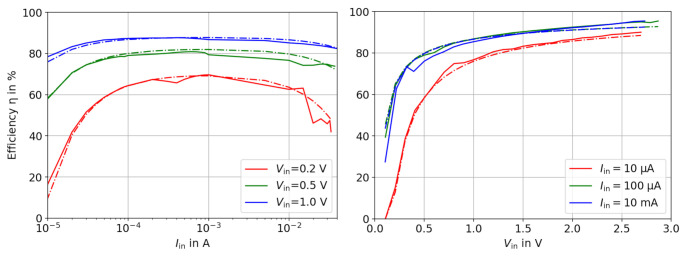
Efficiency w.r.t $I_{\mathrm{in}}$ and $V_{\mathrm{in}}$ of the ADP5090 boost converter. The output voltage $V_{\mathrm{out}}$ is 3.0 V.

**Figure 6 sensors-21-04599-f006:**
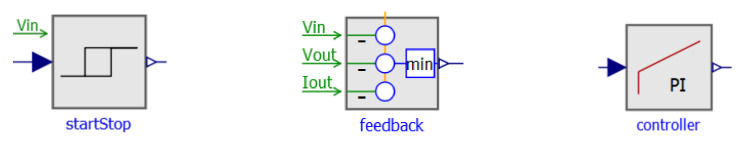
Icons of the components in the control path: start-stop mechanism(**left**), determination of feedback error (**middle**) and the PI controller (**right**).

**Figure 7 sensors-21-04599-f007:**
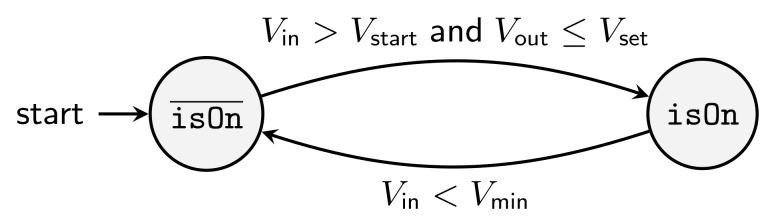
State diagram for isOn to model the start-up and shutdown.

**Figure 8 sensors-21-04599-f008:**
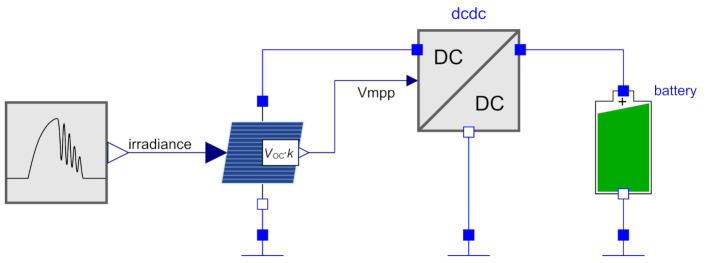
Simulation setup consisting of a solar irradiation simulator, a solar cell and a battery.

**Figure 9 sensors-21-04599-f009:**
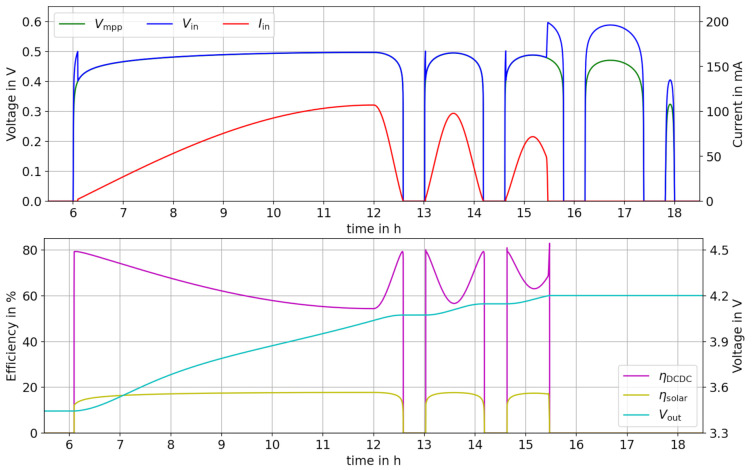
Simulation results showing $I_{\mathrm{in}}$, $V_{\mathrm{in}}$ and $V_{\mathrm{mpp}}$ in the top graph, where $V_{\mathrm{in}}$ follows most of the time $V_{\mathrm{mpp}}$ (covered by blue line). In the bottom graph, the efficiency of the DC/DC converter $\eta_{\mathrm{DCDC}}$ and of the solar cell $\eta_{\mathrm{sol}}$ is shown besides the battery voltage $V_{\mathrm{out}}$, which is increasing monotonously.

**Table 1 sensors-21-04599-t001:** Comparison of state-of-the-art models of behavioral DC/DC converters.

Authors	Electrical Representation	Input Behavior and Start-Up	Feedback Control	Efficiency	Operation at MPP
Torrey et al. [12]	In: current sink Out: current source	$V_{\min}$; but no start-up	commanded $V_{\mathrm{out}}$; limited $I_{\mathrm{out}}$	const. value + input resistor	n/a
Brkic et al. [13]	In: voltage source Out: current source	not modeled	commanded $V_{\mathrm{in}}$	const., 100%	by external vDCRef
Haumer et al. [14]	In: current sink Out: voltage source	not modeled	commanded $V_{\mathrm{out}}$	const., 100%	n/a
Oliver et al. [15]	In: current sink Out: voltage source	remote on/off by state diagram	n/a	look-up table	n/a
Behrmann et al. [16]	In/Out: generic power ports	not modeled	n/a	look-up table	n/a
**this work**	In: conductance Out: current source	$V_{\min}$ and $V_{\mathrm{start}}$	commanded $V_{\mathrm{mpp}}$, $V_{\mathrm{out}}$, $I_{\mathrm{out}}$	function based on power losses	by external $V_{\mathrm{mpp}}$

## Data Availability

The raw data of graphs and tables presented in this study are available on request from the corresponding author.

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
