# Peer review of "Behavioral Modeling of DC/DC Converters in Self-Powered Sensor Systems with Modelica"

_sensors, 2021, doi:10.3390/s21134599_

Round 1

Reviewer 1 Report

Paper presents an interesting study but some improvements such as follows are recommended:

  • in lines 172-173, please revise "Power losses originate from capacative switching losses, the switch resistance and parasitic resistance in the inductor and capacitors [18,19]." for misspellings.
  • conclusions, novelty and main contributions of the paper should be more properly highlighted.

Author Response

Paper presents an interesting study but some improvements such as follows are recommended […].

Thank you very much for reviewing the manuscript and for making the suggested improvements.

In lines 172-173, please revise "Power losses originate from capacative switching losses, the switch resistance and parasitic resistance in the inductor and capacitors [18,19]." for misspellings.

We corrected the word “capacitive”, but we couldn’t find any more errors.

Conclusions, novelty and main contributions of the paper should be more properly highlighted.

We rewrote the section “9 Conclusions” and stated the model highlights as a compact list and explained better our main contribution.

Reviewer 2 Report

In this work the authors simulate a low power consumption DC-DC converter for energy harvester powered system using Modelica language. After describing the various equations and components they show performance of the DC-DC converter when powered by a solar cell.

The study is well conducted and the paper is well written. It may be published after addressing these minor comments and corrections.

  1. In Figure 2 please indicate I_in and V_mpp for clarity. Based on equations 7-10, the entry into efficiency block should be I_in instead of I_out. Please define what is the symbol 'n' at the bottom of Figure 2.
  2. In Figure 4 please indicate the y-axis as efficiency \eta
  3. In equations 7-10, the values of k1 to k4 could change depending on the buck/boost converter chip. This caveat should be mentioned in the manuscript.
  4. Comparing Figure 1 and Figure 6, this work only simulates "power converter 1". What about simulation of "power converter 2"? How would variable input from an electromagnetic or piezoelectric energy harvester change the simulation?

Author Response

In this work the authors simulate a low power consumption DC-DC converter for energy harvester powered system using Modelica language. After describing the various equations and components they show performance of the DC-DC converter when powered by a solar cell.
The study is well conducted and the paper is well written. It may be published after addressing these minor comments and corrections.
Thank you very much for reviewing the manuscript and for making the suggested improvements.

In Figure 2 please indicate I_in and V_mpp for clarity. Based on equations 7-10, the entry into efficiency block should be I_in instead of I_out. Please define what is the symbol 'n' at the bottom of Figure 2.
Thanks for mentioning the typo regarding I_in. The n is the Modelica standard name for a negative analog pin. We added a blue text “(ground)” for a better understanding.

In Figure 4 please indicate the y-axis as efficiency \eta
We added the label to the left-hand side and converted the values to percent (%). Furthermore, we added a grid to comply with Fig 7.

In equations 7-10, the values of k1 to k4 could change depending on the buck/boost converter chip. This caveat should be mentioned in the manuscript.
Good point. We added that caveat to the paragraph. We also added to the conclusion, that more records of other chips are available in the library: “The published library (available online on github [21]) provides datasets for popular PMICs such as the ADP5090 and the BQ25570 and a Python program to fit the parameters.”

Comparing Figure 1 and Figure 6, this work only simulates "power converter 1". What about simulation of "power converter 2"?
The second power converter is typically an LDO or buck-converter, which will operate in the constant-voltage mode. This mode of operation is already covered by the CCCV scheme of the battery. We omitted it to keep the setup clear.

We added an explanatory statement to the paragraph “6. Simulation setup” to the paper.

How would variable input from an electromagnetic or piezoelectric energy harvester change the simulation?
For electromagnetic energy harvesting, a rectifier stage and a buffer capacitor is needed. For piezoelectric energy harvesting, synchronous charge extracting schemes exist. Both topics are beyond of the scope of this paper.
However, we added a new paragraph with an outlook handling these topics to the end of the conclusion.

Reviewer 3 Report

It is interesting paper presenting behavioral model of DC/DC converter.

Please see below minor comments.

  1. Could you provide measurement data to validate or verify the model?
  2. Could you discuss any reasons causing overshoot shown in Fig. 7?
  3. Could you discuss why Modelica is used in this study?

Author Response

It is an interesting paper presenting behavioral model of DC/DC converter.

Thank you very much for reviewing the manuscript and for making the suggested improvements.

Could you provide measurement data to validate or verify the model?

The system simulation (Fig. 7+8) can be carried out by downloading OpenModica and simulating the Model EnergyHarvestingWSN.PowerConverter.UnitTests.SolarADP5090.

We add the path of the model to the paragraph.

For the efficiency fit of the converter, we added all files to the github repository (raw-files and python scripts) and also mentioned this fact in the paper.

Could you discuss any reasons causing overshoot shown in Fig. 7?

Actually, this is not an overshoot, but the transition between a switched-off and switch-on converter. Only a switched-on converter will draw some current from the solar cell and Vin is regulated down to Vmpp.

We rewrote the paragraph.

Could you discuss why Modelica is used in this study?

For us, the equation-based approach was decisive. We rewrote the last part of Sec. 1.2 to highlight the features of Modelica better and added a personal statement to the conclusion.